# Predictors of school students' leisure-time physical activity: An extended trans-contextual model using Bayesian path analysis

Juho Polet[1], Jekaterina Schneider[1], Mary Hassandra[2], Taru Lintunen[1], Arto Laukkanen[1], Nelli Hankonen[3], Mirja Hirvensalo[1], Tuija H. Tammelin[4], Kyra Hamilton[5], Martin S. Hagger[1,6]*

1 Faculty of Sport and Health Sciences, University of Jyväskylä, Jyväskylä, Finland, 2 Department of Physical Education and Sport Science, University of Thessaly, Trikala, Greece, 3 Faculty of Social Sciences, University of Helsinki, Helsinki, Finland, 4 LIKES Research Centre for Physical Activity and Health, Jyväskylä, Finland, 5 School of Applied Psychology, Griffith University, Brisbane, Queensland, Australia, 6 Psychological Sciences, University of California, Merced, Merced, CA, United States of America

* mhagger@ucmerced.edu

## Abstract

The present study aimed to examine effects of motivational and social cognition constructs on children's leisure-time physical activity participation alongside constructs representing implicit processes using an extended trans-contextual model. The study adopted a correlational prospective design. Secondary-school students ($N$ = 502) completed self-report measures of perceived autonomy support from physical education (PE) teachers, autonomous motivation in PE and leisure-time contexts, and social cognition constructs (attitudes, subjective norms, perceived behavioral control), intentions, trait self-control, habits, and past behavior in a leisure-time physical activity context. Five weeks later, students ($N$ = 298) self-reported their leisure-time physical activity participation. Bayesian path analyses supported two key premises of the model: perceived autonomy support was related to autonomous motivation in PE, and autonomous motivation in PE was related to autonomous motivation in leisure time. Indirect effects indicated that both forms of autonomous motivation were related to social cognition constructs and intentions. However, intention was not related to leisure-time physical activity participation, so model variables reflecting motivational processes did not account for substantive variance in physical activity participation. Self-control, attitudes, and past behavior were direct predictors of intentions and leisure-time physical activity participation. There were indirect effects of autonomous motivation in leisure time on intentions and physical activity participation mediated by self-control. Specifying informative priors for key model relations using Bayesian analysis yielded greater precision for some model effects. Findings raise some questions on the predictive validity of constructs from the original trans-contextual model in the current sample, but highlight the value of extending the model to incorporate additional constructs representing non-conscious processes.

**Data Availability Statement:** Data, analysis scripts and output, study materials, and supplemental

materials are available on the Open Science Framework (https://osf.io/z8axj/).

**Funding:** This project was supported by the Finnish Ministry of Education and Culture under grant number OKM/62/626/2016) awarded to MSH and TL; and a Finland Distinguished Professor (FiDiPro) award from Business Finland under grant number 1801/31/2105 awarded to MSH. The funders had no role in study design, data collection and analysis, decision to publish, or preparation of the manuscript.

**Competing interests:** The authors have declared that no competing interests exist.

# Introduction

Research indicates that low levels of physical activity have deleterious effects on the health of young people [1]. However, children and adolescents in many nations are not sufficiently active to confer health benefits and reduce disease risk [2]. As a consequence, national and international health organizations have developed guidelines and national strategies aimed at promoting physical activity in this population [3]. Given the imperative for promoting physical activity among young people, public health organizations and educators have sought to identify optimally effective strategies to enhance physical activity in this population, and the contexts in which these strategies will have maximal reach.

Physical education (PE) has been suggested as a potentially useful existing network that can be utilized to deliver interventions promoting physical activity both inside school, and, importantly, outside school, in children and adolescents [4]. Researchers have, therefore, aimed to explore the potentially efficacious strategies delivered in PE to promote increased physical activity outside of school. Such an endeavor necessitates an understanding of the determinants of children and adolescents' physical activity participation in a PE context and, importantly, whether those determinants relate to physical activity participation outside of school in students' leisure time [5]. Understanding how factors linked to engagement in physical activity in school relate to physical activity performed in another context, leisure time, is critical to informing potential strategies delivered in PE that promote physical activity participation in children and adolescents in their leisure time. Such an approach is also consistent with one of the key pedagogical aims of PE to provide young people with the necessary skills to lead an active lifestyle [6].

## The trans-contextual model

The trans-contextual model [5] was developed to provide a theoretical explanation of the constructs and associated processes that link engagement in physical activity in school PE with physical activity participation in leisure time. Specifically, the model draws on multiple theories to outline relations between school students' motivation toward physical activity in PE and their motives and beliefs toward, and actual participation in, physical activity in their leisure time. The model integrates core constructs and processes from self-determination theory [7], the hierarchical model of intrinsic and extrinsic motivation [8], and the theory of planned behavior [9]. Next, we outline the key premises of the model relating to the determinants of children and adolescents' leisure-time physical activity and the processes involved.

Based on self-determination theory, the first premise of the trans-contextual model focuses on the origins of school students' motivation toward activities in PE, and how their motivation relates to their behavior in PE. The model predicts that the social environment in educational settings fostered by social agents and leaders (e.g., PE teachers) will determine the type or *form* of motivation students experience when performing tasks (e.g., physical activities in PE) and, importantly, their persistence on tasks. Central to the theory is the distinction between autonomous and controlled forms of motivation [10]. Autonomous motivation is a form of motivation reflecting self-endorsed reasons for acting and autonomously motivated individuals tend to persist on tasks and exhibit behavioral persistence. Fostering autonomous motivation through the display of autonomy-supportive behaviors by social agents such as PE teachers will promote autonomous motivation toward physical activities performed in PE. Students who perceive their PE teacher as displaying behaviors that support their autonomy are more likely to report autonomous motivation toward activities in PE [11]. This prediction forms the basis of the first premise of the trans-contextual model: students' perceived autonomy support from their teachers in PE will relate to their autonomous motivation toward physical activity in a PE context.

A central prediction of the trans-contextual model is that there will be a trans-contextual relationship between students' autonomous motivation toward physical activities across PE and leisure-time contexts. This prediction is based on Vallerand's [8] hierarchical model, which describes the process by which motivation is transferred across contexts. Vallerand proposed that individuals experiencing autonomous motivation toward activities in one context will also cite autonomous motives toward similar behaviors in other related contexts. This forms the second premise of the trans-contextual model: school students' level of autonomous motivation toward physical activities in a PE context will be related to their autonomous motivation toward physical activities performed outside of school in their leisure time.

A final prediction of the trans-contextual model is that autonomous motivation toward physical activities in a leisure-time context will be related to students' beliefs and intentions toward, and future participation in, leisure-time physical activity. If an individual has experienced a behavior as autonomously motivated, it is likely to be internalized and integrated into the individual's repertoire of behaviors that satisfy their psychological need for autonomy. They are therefore more likely to actively seek out opportunities to engage in the behavior in future. To do so, they need to align their system of beliefs and intentions involved in the decision to perform that behavior in future. In the trans-contextual model, this process is represented by associations between forms of motivation from self-determination theory and the sets of beliefs from the theory of planned behavior, a leading social cognition theory [9]. Autonomously-motivated individuals are likely to form intentions to perform the behavior in future, and report favorable attitudes, subjective norms, and perceived behavioral control, the immediate belief-based determinants of intentions [12–14]. This forms the third premise of the trans-contextual model: students' autonomous motivation toward physical activity in leisure time will be related to their future participation in leisure-time physical activity mediated by the belief-based social cognition determinants (attitudes, subjective norms, and perceived behavioral control) and their intentions toward participating in leisure-time physical activity in future.

The key premises of the trans-contextual model have received substantial empirical support [5, 15–17]. Furthermore, a meta-analysis of studies applying the model in PE and leisure-time physical activity contexts provides converging evidence supporting model predictions across multiple studies [18].

## Extending the model

While the trans-contextual model has displayed utility in identifying the determinants of leisure-time physical activity participation, it is not without limitations. One limitation is the exclusive focus on motivational and social cognition determinants of leisure-time physical activity participation without regard for the influence of implicit beliefs and motives that affect individuals' behavior beyond their awareness [19]. In contrast, dual-process theories of motivation and social cognition propose that individuals' behavior is determined by constructs that reflect conscious, reasoned decision making (e.g., autonomous motivation, social cognition constructs) but also by constructs that reflect implicit decision-making that impact behavior with little reasoned deliberation. Such constructs include implicit attitudes, habits, and individual difference constructs [20–23]. Non-conscious processes are adaptive because they lead to effective, efficient decision-making when reasoned deliberation is unnecessary or particularly costly [24]. Constructs reflecting these non-conscious processes are proposed to impact behavior directly without mediation by intentions, independent of the reasoned processes. These constructs may, therefore, account for the additional variance in leisure-time physical activity participation in the trans-contextual model and serve to provide a more comprehensive prediction of physical activity motivation and behavior in PE and leisure time.

Prominent behavioral determinants that reflect non-conscious processes are habit, trait self-control, and affective attitudes. Focusing on habit, although research has historically considered effects of past behavior as a viable proxy for habitual effects [25], recent research has focused on habit as a psychological construct [21, 26]. Theories of habit suggest that habitual behaviors are a function of behavioral experiences in the presence of consistent environmental, situational, or internal cues, and are often experienced as automatic, effortless, and highly accessible [26]. These components have been captured by self-report measures of habit, which are meta-cognitive measures which tap individuals' experience of target behaviors as 'unthinking' and 'automatic' [21]. Such measures have been shown to predict behavior independent of intention-mediated measures [27–30], and are also associated with action accessibility and behavioral performance in stable contexts [31, 32].

While habits are expected to predict intentions and behavior, they may also be related to motives that form part of the 'motivational sequence' proposed in the trans-contextual model. Individuals that hold autonomous motives toward behaviors like leisure-time physical activity, are more likely to persist with those behaviors, because the behavior fulfils psychological needs and leads to adaptive outcomes in the absence of external contingencies. This has been supported in the research literature on self-determination theory with consistent links between autonomous motivation and physical activity [33]. Related to this, autonomously motivated individuals are also more likely to form habits for those behaviors because they are likely to perform those behaviors in a consistent fashion with high frequency and in stable contexts–the key conditions under which habits form [34, 35]. This has been supported by previous research on the autonomous motivation-habit relationship [36]. Based on these premises, it follows that autonomous motivation toward physical activity in PE and leisure-time contexts is expected to be indirectly related to intentions and physical activity behavior mediated by habits. This proposed effect will be independent of the 'motivational sequence' proposed in the model, consistent with the dual-process approaches outlined previously.

Trait self-control reflects individual differences in capacities and self-regulatory skills that enable individuals to resist impulses and temptations, and engage in sustained, effortful behavior to attain long-term goal-directed outcomes [37]. Trait self-control has been consistently related to adaptive behaviors, including physical activity, across multiple contexts and populations [38]. Research has also demonstrated that behavioral effects of trait self-control may be direct, independent of intentions [39]. Such effects reflect generalized tendencies to engage in adaptive behaviors without the need for deliberation or consideration. However, a case has also been made for effects of trait self-control on behavior mediated by intentions [39]. Such effects reflect situations where individuals have to actively engage in effortful deliberation to overcome a maladaptive behavior, or engage in a new behavior, that requires deliberation. Effects of trait self-control in motivational and social cognition theories may, therefore, relate to behavior via two pathways, directly, and indirectly through intentions. Research incorporating trait self-control in the model has supported these dual effects, with direct and intention-mediated effects on physical activity participation [39].

While self-control has been identified as an independent determinant of intentions and behavior, there is also research that has linked self-control, and self-regulatory processes in general, with the forms of motivation implicated in the motivational sequence of the trans-contextual model. For example, research has highlighted that individuals reporting self-determined motives are less likely to be vulnerable to self-control failure and ego-depletion [40–42], and more likely to report intentions toward, and participate in, future behaviors, including physical activity [36, 39, 43]. These findings are consistent with the self-determination theory hypothesis that autonomous motivation is 'energizing' and individuals with autonomous motives toward behaviors are likely to report greater capacity to perform the behavior, and

hence greater self-control [36]. This hypothesis also aligns with the premise of the trans-contextual model that autonomous motives lead individuals to mobilize their resources to perform need-satisfying behaviors in future. Consistent with these proposals, it is reasonable to expect that autonomous motivation will be indirectly related to leisure-time physical activity intentions and behavior through self-control.

There is also research demonstrating that attitudes may predict behavior directly, and such direct effects may also reflect non-conscious decision making [44, 45]. The original conceptualization of the theory of planned behavior specifies that attitudes represent cognitive reflections on future participation in a target behavior and should relate to behavior mediated by intentions. However, direct effects of attitudes have been identified [45], and have been attributed to the affective or *emotional* component of attitude. Research separating the cognitive and affective attitude components has demonstrated independent effects. The affective component is proposed to encompass visceral approach or avoidance responses learned through behavioral experience [44]. Direct effects of attitude on behavior may reflect a further spontaneous, automatic process, which affects behavior independent of intentions.

## The present study

In the present study, we aimed to extend the trans-contextual model by including self-control, habits, and attitudes as additional direct determinants of physical activity intentions and behavior participation. This extension is expected to provide additional information on the determinants of leisure-time physical activity behavior, particularly effects of constructs representing non-conscious processes not accounted for in the original model. We also applied the Bayesian analytic approach to test model effects using informative prior values for key model effects derived from a meta-analysis of the model [18]. We also capitalized on previous research on self-reported habit [36, 46] and trait self-control [38] to specify informative priors for these parameters in our test of extensions to the model. We expected to see a reduced level of uncertainty in the distributions of the parameters of the model specified for the current data when informative priors for key model parameters derived from the meta-analysis are specified, reflected in narrowed credibility intervals, compared to the distributions when non-informative priors are specified.

Specifically, the study adopted survey methods and a five-week prospective design with measures of motivational and social cognition constructs, habit, trait-self-control, and past physical activity participation taken at an initial occasion, and self-reported leisure-time physical activity participation taken at follow-up, five weeks later. This time period was selected to provide reasonable medium-term prediction, which exceeds typical time frames in model tests [47]. In addition to testing effects of the motivational and social cognition determinants from the trans-contextual model on students' intentions toward, and actual participation in, leisure-time physical activity, we also tested direct effects of the constructs reflecting non-conscious processes as direct determinants of leisure-time physical activity participation: self-reported habit, trait self-control, and attitudes. Further, effects of autonomous motivation in both contexts were expected to be indirectly related to physical activity intentions and behavior in leisure time mediated by both habit and self-control. Such effects represent processes by which self-determined motivation promotes behavioral enactment by promoting greater perceived self-regulatory capacity [42] and experience of the behavior as automatic [48]. Finally, we also expected model effects to hold in the presence of past behavior, and that there would be an indirect effect of past physical activity behavior on leisure-time physical activity mediated by habit. Such a relationship would illustrate the extent to which past behavior is a function of habit formation, consistent with previous theory and research [27, 48]. The specific predictions

of the proposed model, including direct and indirect effects in the proposed model are summarized in Table 1 and Fig 1.

## Method

### Participants

Participants were lower secondary school students ($N$ = 502, 43.82% female; $M$ age = 14.52, $SD$ = 0.71) recruited from selected schools across Jyväskylä, Finland with support from the City Education Department. The University institutional review board and Education Department approved the study protocol prior to data collection. Informed consent was sought from the head teacher of each school, and, subsequently, PE teachers and eligible students' parents or legal guardians via the schools' online administration and communication software or via email or post. Opt-in consent was sought from the head teachers and PE teachers, while opt-out consent was sought from students' parents and legal guardians. Qualified full-time PE teachers teaching regular PE lessons in lower secondary schools were eligible to participate in the study and were asked to select one of their PE classes to take part. Students in grades 7 to 9 (typical ages 13 to 15 years) in lower secondary schools were eligible to participate. Students with existing physical or mental health conditions that prevented participation in PE lessons, regular leisure-time physical activity, or completing surveys were excluded.

### Design and procedure

Data for this study was collected as part of a larger randomized controlled trial (trial registration: ISRCTN39374060; PETALS). The trial adopted a cluster-randomized, waitlist control, single-group intervention design with randomization by school. The trial comprised a teacher training phase and an implementation phase; full details of the intervention design and content have been published previously [50]. Secondary school PE teachers ($N$ = 29) from 11 secondary schools and their students ($N$ = 502) were invited to participate in the study. The pool of potentially eligible students numbered approximately 5000 across the 11 schools. Baseline data was collected prior to the teacher training phase and participants completed self-report questionnaires assessing demographic, psychological, and behavioral measures. The baseline data collection period was followed by the teacher training phase (12 hours over two weeks) and the implementation phase (one month), after which post-intervention data was collected comprising the same self-report questionnaires as at baseline. Follow-up data was further collected one, three, and six months post-intervention. The present study used measures of motivation and social cognition constructs and leisure-time physical activity participation taken at baseline and post-intervention leisure-time physical activity participation controlling intervention effects at baseline. Data for the present study were collected between September and December 2018.

### Measures

Measures of study constructs were adapted from instruments used in previous applications of the trans-contextual model. Measures included in the surveys were: perceived autonomy support from PE teachers [51]; autonomous motivation derived from items measuring self-determined forms of motivation from the perceived locus of causality scales for the PE and leisure-time physical activity contexts [52]; intentions, attitudes, subjective norms, and perceived behavioral control from the theory of planned behavior [53]; self-reported habit [21] and trait self-control [54]; and self-reported leisure-time physical activity participation [55]. All self-report measures were previously translated from English to Finnish using a back-translation process by two bilingual researchers. All measures used in the current research exhibited acceptable construct validity in

**Table 1. Summary of hypothesized direct and indirect effects in the extended trans-contextual model.**

| Hypothesis (H) | Independent variable | Dependent variable | Mediator(s) | Informative priors | |
|---|---|---|---|---|---|
| | | | | β | σ² |
| Direct effects | | | | | |
| H₁ | PAS | Aut. mot. (PE)[a] | – | 0.42 | 0.10 |
| H₂ | Aut. mot. (PE) | Aut. mot. (LT)[a] | – | 0.56 | 0.17 |
| H₃ | PAS | Aut. mot. (LT)[a] | – | 0.29 | 0.18 |
| H₄ | Aut. mot. (LT) | Attitude[a] | – | 0.60 | 0.12 |
| H₅ | Aut. mot. (LT) | Subjective norm[a] | – | 0.26 | 0.26 |
| H₆ | Aut. mot. (LT) | PBC[a] | – | 0.51 | 0.19 |
| H₇ | Aut. mot. (LT) | Intention[a] | – | 0.31 | 0.13 |
| H₈ | Attitude | Intention[a] | – | 0.68 | 0.09 |
| H₉ | Subjective norm | Intention[a] | – | 0.42 | 0.25 |
| H₁₀ | PBC | Intention[a] | – | 0.63 | 0.28 |
| H₁₁ | Habit | Intention | – | – | – |
| H₁₂ | Self-control | Intention | – | – | – |
| H₁₃ | Intention | Phys. act.[a] | – | 0.60 | 0.20 |
| H₁₄ | Attitude | Phys. act.[a] | – | 0.43 | 0.21 |
| H₁₅ | PBC | Phys. act.[a] | – | 0.43 | 0.21 |
| H₁₆ | Habit | Phys. act.[b] | – | 0.43 | 0.13 |
| H₁₇ | Self-control | Phys. act.[c] | – | 0.26 | 0.09 |
| H₁₈ | Aut. mot. (LT) | Phys. act. | – | – | – |
| H₁₉ | Aut. mot. (LT) | Habit[d] | – | 0.20 | 0.43 |
| H₂₀ | Aut. mot. (LT) | Self-control | – | – | – |
| H₂₁ | Past behavior | Habit | – | – | – |
| H₂₂ | Past behavior | Phys. act. | – | – | – |
| Indirect effects | | | | | |
| H₂₃ | PAS | Aut. mot. (LT) | Aut. mot. (PE) | – | – |
| H₂₄ | Aut. mot. (PE) | Intention | Aut. mot. (LT) | – | – |
| | | | Attitude | | |
| H₂₅ | Aut. mot. (PE) | Intention | Aut. mot. (LT) | – | – |
| | | | Sub. norm. | | |
| H₂₆ | Aut. mot. (PE) | Intention | Aut. mot. (LT) | – | – |
| | | | PBC | | |
| H₂₇ | Aut. mot. (PE) | Phys. act. | Aut. mot. (LT) | – | – |
| | | | Attitude | | |
| | | | Intention | | |
| H₂₈ | Aut. mot. (PE) | Phys. act. | Aut. mot. (LT) | – | – |
| | | | Sub. norm. | | |
| | | | Intention | | |
| H₂₉ | Aut. mot. (PE) | Phys. act. | Aut. mot. (LT) | – | – |
| | | | PBC | | |
| | | | Intention | | |
| H₃₀ | Aut. mot. (PE) | Phys. act. | Aut. mot. (LT) | – | – |
| | | | PBC | | |
| H₃₁ | Aut. mot. (LT) | Intention | Attitude | – | – |
| H₃₂ | Aut. mot. (LT) | Intention | Sub. norm. | – | – |
| H₃₃ | Aut. mot. (LT) | Intention | PBC | – | – |
| H₃₄ | Aut. mot. (LT) | Intention | Habit | – | – |

(*Continued*)

**Table 1.** (Continued)

| Hypothesis (H) | Independent variable | Dependent variable | Mediator(s) | Informative priors | |
|---|---|---|---|---|---|
| | | | | β | σ² |
| $H_{35}$ | Aut. mot. (LT) | Intention | Self-control | – | – |
| $H_{36}$ | Aut. mot. (LT) | Phys. act. | Attitude | – | – |
| | | | Intention | | |
| $H_{37}$ | Aut. mot. (LT) | Phys. act. | Sub. norm. | – | – |
| | | | Intention | | |
| $H_{38}$ | Aut. mot. (LT) | Phys. act. | PBC | – | – |
| | | | Intention | | |
| $H_{39}$ | Aut. mot. (LT) | Habit | Phys. act | – | – |
| $H_{40}$ | Aut. mot. (LT) | Self-control | Phys. act | – | – |
| $H_{41}$ | Past beh. | Habit | Phys. act | – | – |

*Note.* [a]Prior values derived from Hagger and Chatzisarantis [18];

[b]Prior values derived from Gardner et al. [49];

[c]Prior values derived from de Ridder et al. [38];

[d]Prior value derived from Kaushal et al. [36]. PAS = Perceived autonomy support; Aut. mot. = Autonomous motivation; PE = Physical education context; LT = Leisure-time context; PBC = Perceived behavioral control; Sub. norm = Subjective norm; Phys. act = Self-reported leisure-time physical activity participation; Past. beh. = Past leisure-time physical activity behavior.

previous research applying confirmatory factor analyses. Items tapping each construct exhibited satisfactory factor loadings, average variance extracted, and composite reliability estimates [5, 46, 56]. Furthermore, we also conducted a pilot study in which the validity of the translated trans-contextual measures was tested in the target population [57]. This study used single-indicator latent variable model using omega reliability estimates to control for measurement error. Simulation research using full structural equation models and single-indicator models has revealed little difference in the parameter estimates in models either approach [58]. These data provided support for the use of these measures in this population, particularly the construct and predictive validity of the measures in the context of the trans-contextual model. Full details of the measures used are available in Appendix A in S1 File.

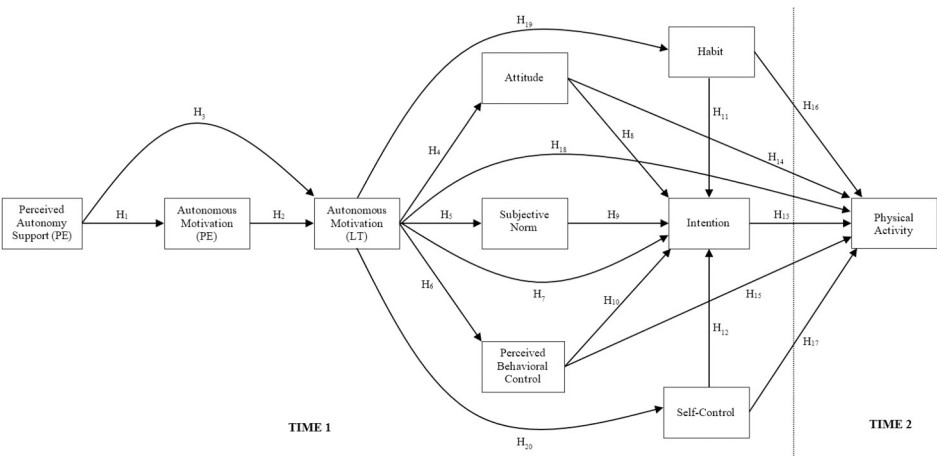

**Fig 1. Hypothesized relations among constructs of the extended trans-contextual model.** Effects of past behavior on all other model constructs have been omitted for clarity.

**Perceived autonomy support.** Students' perceived autonomy support from their PE teacher was measured using items from the Perceived Autonomy Support Scale for Exercise Settings [51]. The scale comprised 13 items (e.g., "I feel that my PE teacher provides me with choices and options to . . .") with responses provided on seven-point scales (1 = *strongly disagree* and 7 = *strongly agree*).

**Autonomous motivation.** Autonomous motivation toward in-school and out-of-school physical activities was measured using items from the Perceived Locus of Causality Questionnaire [52]. Two items measured *identified regulation* (e.g., "I do PE/physical activity because it is important to me to do well in PE/physical activity") and two items measured *intrinsic motivation* (e.g., "I do PE/physical activity because it is fun"). Responses were provided on seven-point scales (1 = *not true for me* and 7 = *very true for me*). For each of the PE and leisure-time contexts, a composite autonomous motivation score was computed by averaging scores on the identified regulation and intrinsic motivation items.

**Theory of planned behavior constructs.** Measures of students' attitudes, subjective norms, perceived behavioral control, and intentions with respect to their future participation in leisure-time physical activity were developed according to guidelines [53]. Attitudes were measured on three items in response to a common stem: "Participating in physical activity in the next five weeks will be. . .", with responses provided on seven-point scales (e.g., 1 = *unenjoyable* and 7 = *enjoyable*). Subjective norms (e.g., "Most people who are important to me think I should do active sports and/or vigorous physical activities during my leisure time in the next five weeks"), perceived behavioral control (e.g., "I am confident I could do active sports and/or vigorous physical activities during my leisure time in the next five weeks"), and intentions (e.g., "I intend to do active sports and/or vigorous physical activities during my leisure time in the next five weeks") were measured using two items each with responses provided on seven-point scales (e.g., 1 = *strongly disagree* and 7 = *strongly agree*).

**Habit.** Habit was measured using automaticity items from the Self-Reported Habit Index [21] which focuses on personal experience of the behavior as 'automatic' and excludes items related to past behavior. The scale comprised four items (e.g., "Physical activity is something I do without thinking") with responses provided on seven-point scales (1 = *completely disagree* and 7 = *completely agree*).

**Trait self-control.** Students' trait self-control was measured using the 10-item Self-Discipline Scale (e.g., "I tend to carry out my plans") from the IPIP-HEXACO scales [59] with responses provided on four-point scales (1 = *not like me at all* and 4 = *very much like me*).

**Behavior.** Past leisure-time physical activity at baseline and leisure-time physical activity participation at follow-up was measured using the short form of the International Physical Activity Questionnaire (IPAQ; 55). The IPAQ comprises four items recording the frequency (number of days) and duration (minutes per day) of engagement in moderate and vigorous physical activity, walking, and sitting over the past seven days. IPAQ data were processed according to established guidelines [60]. The procedure gives an estimate of physical activity in MET-minutes per week with higher MET-minute values indicating higher level of physical activity engagement. Full details of calculations used to produce physical activity estimates are presented in Appendix B in S1 File. Internal consistency values for the short-form IPAQ exceed guideline cut-off values and scores on the scale demonstrate reasonable agreement with the long form in previous research [55].

## Data analysis

The proposed hypotheses of the extended trans-contextual model (see Table 1 and Fig 1) were tested using Bayesian path analytic models estimated with the Mplus 7.31 statistical software.

We computed composite scales study measures by computing an average of the items for each construct. We controlled for the effects of the intervention in the model by including effects of a binary variable representing intervention group membership (1 = allocated autonomy support intervention group, 0 = allocated to control group) on follow-up leisure-time physical activity participation. We controlled for effects of gender as a binary variable (0 = female, 1 = male) and age as a continuous variable by estimating effects of these variables on all other constructs in the model. Missing data for the model components were imputed using full information maximum likelihood (FIML) in Mplus.

A Markov Chain Monte Carlo (MCMC) simulation process using Gibbs' algorithm [61] was applied to estimate our Bayesian path models in the present study. The first 50% of the iterations ($N$ = 100,000) was used as a burn-in phase with the remainder used to test the specified model parameters. We established convergence of the parameter estimates in the Bayesian models according to the Gelman-Rubin [62] criterion based on a potential scale reduction (PSR) value of 1.01. Our analysis required estimation of two models. We first estimated a model that adopted the non-informative default priors available in Mplus to estimate model parameters. The defaults assumed a normal distribution with a mean of zero and a variance value essentially equivalent to infinity ($10^{10}$). In our second model, we applied informative prior values taken from previous research [18, 36, 38, 49] to estimate model parameters. Specifically, priors for relations among the trans-contextual model constructs were taken from Hagger and Chatzisarantis' [18] meta-analysis. The prior value for the effect of self-reported habit on leisure-time physical activity participation was derived from Gardner et al.'s [46] meta-analysis of self-reported habit in physical activity. The prior value for the effect of trait self-control on leisure-time physical activity participation was taken from de Ridder et al.'s [38] meta-analysis of trait self-control in health behaviors. The prior value for the effect of autonomous motivation in leisure time on habit was taken from Kaushal et al.'s [36] integrated model test. Finally, we used non-informative prior values for remaining parameters without user-specified priors.

We used multiple published criteria to establish the goodness-of-fit of our proposed models with the data across the iterations of the Bayesian analysis [61]. These criteria included the 95% confidence interval of the difference in the goodness-of-fit chi-square value across the observed and replicated models, as well as the posterior predictive *p*-value (PPP). The goodness-of-fit chi-square value should have a positive upper limit, a negative lower limit, and be centered about zero, and the PPP value should be greater than .05 and approach .50, for well-fitting models. In addition, we also report the Bayesian Information Criterion (BIC), a relative fit index which allows researchers to identify the model with the greatest parsimony relative to fit as it includes a term that 'penalizes' overfitting. Cut-off values of .95 or greater for the CFI and TLI, and .06 for the RMSEA, have been proposed as indicative of good model fit. Furthermore, the 90% confidence intervals of each index should ideally exceed the cut-off values.

With respect to model parameters, a point estimate and 95% posterior credibility interval was produced for each parameter in the models. A non-zero credibility interval for a parameter provides confirmatory support for the proposed effect in the model. Point estimates and credibility intervals were also produced for the proposed indirect effects in the models [63]. In addition, we also expected that specification of informative prior values in the second model would lead to increased precision in the point estimates. To demonstrate changes in precision, we followed Yuan and McKinnon's method which evaluates the extent to which the posterior credibility intervals about each parameter estimate is narrowed with the introduction of informative priors. If the width of the confidence intervals is reduced, we have sharp confirmation that inclusion of the prior values alongside the current data leads to increased precision in estimates. Data, syntax, and output files for our analyses are available online: https://osf.io/z8axj.

## Results

### Preliminary analyses

Of the participants who completed the initial survey ($N$ = 502), 370 provided complete data at baseline and 298 participants (50% female; $M$ age = 14.51, $SD$ = 0.70) provided data for analysis after the second survey (19.46% attrition rate). Attrition was due to school absences. Percentage of missing data for the psychological constructs over time was low ($M$ = 1.4%; range = 0.0% to 4.0%) and Little's MCAR test ($\chi^2$ = 63.882, df = 70, $p$ = .683) suggested the data were missing at random. In addition, there were no significant differences between those who completed study measures at both time points and those who did not on gender distribution, ($\chi^2(1)$ = 0.403, $p$ = .526), age ($t(368)$ = -0.463, $p$ = .643), or baseline physical activity ($t(368)$ = -1.103, $p$ = .271). We also conducted a one-way MANOVA to examine differences in psychological variables between those who completed study measures at both time points and those who did not, which was not significant ($F(9, 360)$ = 0.775, $p$ = .639; Wilks' $\Lambda$ = .981; partial $\eta^2$ = .019). All constructs exhibited adequate internal consistency. Means, standard deviations, omega internal consistency coefficients and zero-order intercorrelations among study constructs are presented in Appendix C in S1 File.

### Path analyses

Bayesian path analytic models using non-informative (Bayesian posterior predictive $\chi^2$ 95% CI = [-26.087, 57.542]; PPP = 0.225; BIC = 9202.749) and informative priors for key model relationships (Bayesian posterior predictive $\chi^2$ 95% CI = [-18.211, 67.825]; prior PPP = 0.126; BIC = 9081.715) exhibited adequate goodness-of-fit with the data. In addition, the BIC indicated that the model that included informative priors exhibited better fit than the model with non-informative priors. Parameter estimates and 95% credibility intervals for the analysis with non-informative priors (Model 1) and the analysis including informative priors for key model relationships (Model 2) are presented in Table 2 and Fig 2.

It is important to note that we did not conduct an a priori statistical power analysis for the current study as the main purpose of the trial was to test the effects of the intervention [50]. Nevertheless, we conducted a posteriori power analysis to check whether we had adequate power to test the current model. Our analysis was based on MacCallum, Browne, and Sugawara's [64] statistical power determination based on the RMSEA. The final sample size ($N$ = 298), a null hypothesis RMSEA of 0 and a study hypothesis RMSEA of 0.064, alpha set at 0.05, and 16 degrees of freedom were inputs for the analysis, which was conducted using the Webpower package in R [65]. The resulting power estimate of 0.805 indicated that we had sufficient statistical power to detect effects of the stipulated size.

Assuming the selected priors derived from meta-analyses were indicative of the population point estimates and distributions of effects among model constructs, we expected that Model 2 would yield greater precision in model parameter estimates compared to Model 1 [63]. This was evaluated by examining the extent to which the credibility intervals about each parameter estimate differed across the models (Table 2). Results indicated that the width of the credibility intervals was narrowed for a few of the effects in Model 2 relative to Model 1, but not by a substantial margin in most cases. The adequate fit of both models suggests that including informative priors in the analyses for key model relationships did not have a substantial bearing on the pattern of effects in the model. Nevertheless, given that some parameters were more precise, particularly the direct effects of perceived autonomy support on autonomous motivation in PE and autonomous motivation in PE on autonomous motivation in leisure time, and the indirect effects of autonomous motivation in PE on autonomous motivation in leisure time and

**Table 2. Parameter estimates (β) with 95% credibility intervals for hypothesized effects from the Bayesian path analyses of the extended trans-contextual model for leisure-time physical activity.**

| H | Independent variable | Dependent variable | Mediator(s) | Model 1 | | | Model 2 | | | %diff |
|---|---|---|---|---|---|---|---|---|---|---|
| | | | | β | 95% CrI | | β | 95% CrI | | |
| | | | | | LL | UL | | LL | UL | |
| Direct effects | | | | | | | | | | |
| $H_1$ | PAS | Aut. mot. (PE)† | – | 2.403*** | 1.485 | 3.018 | 1.222*** | 0.625 | 1.634 | -34.18 |
| $H_2$ | Aut. mot. (PE) | Aut. mot. (LT)† | – | 0.449*** | 0.364 | 0.536 | 0.452*** | 0.366 | 0.537 | -0.58 |
| $H_3$ | PAS | Aut. mot. (LT)† | – | 0.140* | 0.012 | 0.264 | 0.138* | 0.011 | 0.265 | 0.79 |
| $H_4$ | Aut. mot. (LT) | Attitude† | – | 0.433*** | 0.343 | 0.523 | 0.437*** | 0.348 | 0.526 | -1.11 |
| $H_5$ | Aut. mot. (LT) | Sub. norm† | – | 0.164* | 0.007 | 0.321 | 0.166* | 0.012 | 0.322 | -1.27 |
| $H_6$ | Aut. mot. (LT) | PBC† | – | 0.221*** | 0.119 | 0.322 | 0.225*** | 0.125 | 0.326 | -0.99 |
| $H_7$ | Aut. mot. (LT) | Intention† | – | 0.410*** | 0.304 | 0.516 | 0.403*** | 0.298 | 0.507 | -1.42 |
| $H_8$ | Attitude | Intention† | – | 0.166** | 0.057 | 0.275 | 0.182*** | 0.076 | 0.289 | -2.29 |
| $H_9$ | Sub. norm | Intention† | – | 0.118*** | 0.058 | 0.178 | 0.118*** | 0.059 | 0.178 | -0.83 |
| $H_{10}$ | PBC | Intention† | – | 0.325*** | 0.227 | 0.422 | 0.324*** | 0.228 | 0.420 | -1.54 |
| $H_{11}$ | Habit | Intention | – | 0.057 | -0.029 | 0.144 | 0.058 | -0.028 | 0.144 | -0.58 |
| $H_{12}$ | Self-control | Intention | – | 0.217* | 0.032 | 0.400 | 0.214* | 0.030 | 0.399 | 0.27 |
| $H_{13}$ | Intention | Phys. act.† | – | -0.001 | -0.062 | 0.060 | 0.000 | -0.060 | 0.061 | -0.82 |
| $H_{14}$ | Attitude | Phys. act.† | – | 0.092** | 0.032 | 0.151 | 0.093** | 0.034 | 0.152 | -0.84 |
| $H_{15}$ | PBC | Phys. act.† | – | -0.029 | -0.085 | 0.026 | -0.029 | -0.084 | 0.026 | -0.90 |
| $H_{16}$ | Habit | Phys. act.† | – | -0.005 | -0.051 | 0.040 | -0.004 | -0.049 | 0.042 | 0.00 |
| $H_{17}$ | Self-control | Phys. act.† | – | 0.104* | 0.003 | 0.206 | 0.107* | 0.007 | 0.207 | -1.48 |
| $H_{18}$ | Aut. mot. (LT) | Phys. act. | – | -0.017 | -0.079 | 0.045 | -0.019 | -0.080 | 0.043 | -0.81 |
| $H_{19}$ | Aut. mot. (LT) | Habit† | – | 0.121 | -1.592 | 2.637 | 0.532*** | 0.392 | 0.671 | -93.40 |
| $H_{20}$ | Aut. mot. (LT) | Self-control | – | 0.108*** | 0.046 | 0.171 | 0.138*** | 0.071 | 0.207 | 8.80 |
| $H_{21}$ | Past behavior | Habit | – | 1.521*** | -3.089 | 0.658 | 0.799*** | 0.397 | 1.206 | -78.41 |
| $H_{22}$ | Past beh. (LT) | Phys. act. | – | 0.675*** | 0.519 | 0.829 | 0.672*** | 0.518 | 0.827 | -0.32 |
| Indirect effects | | | | | | | | | | |
| $H_{23}$ | PAS | Aut. mot. (LT) | Aut. mot. (PE) | 1.059*** | 0.643 | 1.449 | 0.547*** | 0.276 | 0.777 | -37.84 |
| $H_{24}$ | Aut. mot. (PE) | Intention | Aut. mot. (LT) | 0.032** | 0.011 | 0.057 | 0.035*** | 0.014 | 0.061 | 2.17 |
| | | | Attitude | | | | | | | |
| $H_{25}$ | Aut. mot. (PE) | Intention | Aut. mot. (LT) | 0.008* | 0.000 | 0.020 | 0.008* | 0.001 | 0.020 | -5.00 |
| | | | Sub. norm. | | | | | | | |
| $H_{26}$ | Aut. mot. (PE) | Intention | Aut. mot. (LT) | 0.032*** | 0.015 | 0.053 | 0.032*** | 0.016 | 0.054 | 0.00 |
| | | | PBC | | | | | | | |
| $H_{27}$ | Aut. mot. (PE) | Phys. act. | Aut. mot. (LT) | 0.000 | -0.002 | 0.002 | 0.000 | -0.002 | 0.002 | 0.00 |
| | | | Attitude | | | | | | | |
| | | | Intention | | | | | | | |
| $H_{28}$ | Aut. mot. (PE) | Phys. act. | Aut. mot. (LT) | 0.000 | -0.001 | 0.001 | 0.000 | -0.001 | 0.001 | 0.00 |
| | | | Sub. norm. | | | | | | | |
| | | | Intention | | | | | | | |
| $H_{29}$ | Aut. mot. (PE) | Phys. act. | Aut. mot. (LT) | 0.000 | -0.002 | 0.002 | 0.000 | -0.002 | 0.002 | 0.00 |
| | | | PBC | | | | | | | |
| | | | Intention | | | | | | | |
| $H_{30}$ | Aut. mot. (PE) | Phys. act. | Aut. mot. (LT) | -0.003 | -0.009 | 0.003 | -0.003 | -0.009 | 0.003 | 0.00 |
| | | | PBC | | | | | | | |
| $H_{31}$ | Aut. mot. (LT) | Intention | Attitude | 0.071** | 0.024 | 0.123 | 0.079*** | 0.032 | 0.131 | 0.00 |
| $H_{32}$ | Aut. mot. (LT) | Intention | Sub. norm. | 0.018* | 0.001 | 0.043 | 0.019* | 0.001 | 0.044 | 2.38 |

*(Continued)*

**Table 2.** (Continued)

| H | Independent variable | Dependent variable | Mediator(s) | Model 1 | | | Model 2 | | | %diff |
|---|---|---|---|---|---|---|---|---|---|---|
| | | | | β | 95% CrI | | β | 95% CrI | | |
| | | | | | LL | UL | | LL | UL | |
| $H_{33}$ | Aut. mot. (LT) | Intention | PBC | 0.071*** | 0.035 | 0.115 | 0.072*** | 0.037 | 0.115 | -2.50 |
| $H_{34}$ | Aut. mot. (LT) | Intention | Habit | 0.003 | -0.117 | 0.240 | 0.030 | -0.015 | 0.079 | -73.67 |
| $H_{35}$ | Aut. mot. (LT) | Intention | Self-control | 0.022* | 0.003 | 0.052 | 0.028* | 0.004 | 0.064 | 22.45 |
| $H_{36}$ | Aut. mot. (LT) | Phys. act. | Attitude | 0.000 | -0.005 | 0.005 | 0.000 | -0.005 | 0.005 | 0.00 |
| | | | Intention | | | | | | | |
| $H_{37}$ | Aut. mot. (LT) | Phys. act. | Sub. norm. | 0.000 | -0.001 | 0.001 | 0.000 | -0.001 | 0.001 | 0.00 |
| | | | Intention | | | | | | | |
| $H_{38}$ | Aut. mot. (LT) | Phys. act. | PBC | 0.000 | -0.005 | 0.005 | 0.000 | -0.005 | 0.005 | 0.00 |
| | | | Intention | | | | | | | |
| $H_{39}$ | Aut. mot. (LT) | Phys. act | Habit | 0.000 | -0.071 | 0.056 | -0.002 | -0.027 | 0.023 | -60.63 |
| $H_{40}$ | Aut. mot. (LT) | Phys. act | Self-control | 0.011 | 0.000 | 0.026 | 0.014* | 0.001 | 0.033 | 23.08 |
| $H_{41}$ | Past beh. | Phys. act | Habit | -0.002 | -0.127 | 0.108 | -0.003 | -0.042 | 0.035 | -67.23 |

*Note.* †Parameters with informative priors. Model 1 = Bayesian path model with non-informative priors; Model 2 = Bayesian path model including informative priors; H = Hypothesis; β = Parameter estimate; 95% CrI = 95% credibility interval of path coefficient; %diff = Percent difference in 95% credibility interval of path coefficients of path analysis including informative priors for specified model relationships compared to analysis using non-informative priors (negative numbers indicate a narrowing of credibility intervals when using informative priors); PAS = Perceived autonomy support; Aut. mot. = Autonomous motivation; PE = Physical education contexts; LT = Leisure-time context; PBC = Perceived behavioral control; Sub. norm = Subjective norm; Phys. act = Self-reported leisure-time physical activity participation; Past. Beh. = Past leisure-time physical activity behavior.

\* $p < .05$

\*\* $p < .01$

\*\*\* $p < .001$.

intention, we elected to evaluate our hypothesis tests based on the model using informative priors (Model 2). If the posterior distribution for each effect, represented by the credibility intervals about the coefficients, did not include zero, then the effect was considered supported and the posterior probability of a non-zero value for the coefficient exceeds 0.975.

In terms of direct effects, we found a non-zero effect of perceived autonomy support in PE on autonomous motivation in PE ($H_1$; β = 1.222, 95% CI [0.6425, 1.634], $p < .001$). There was also a non-zero trans-contextual effect of autonomous motivation in PE on autonomous motivation in leisure time ($H_2$; β = 0.452, 95% CI [0.366, 0.537], $p < .001$), and perceived autonomy support in PE also had a non-zero effect on autonomous motivation in leisure time ($H_3$; β = 0.138, 95% CI [0.011, 0.265], $p < .001$). We also found non-zero effects of autonomous motivation in leisure time on attitudes ($H_4$; β = 0.437, 95% CI [0.348, 0.526], $p < .001$), subjective norms ($H_5$; β = 0.166, 95% CI [0.012, 0.322], $p = .017$), and perceived behavioral control ($H_6$; β = 0.225, 95% CI [0.125, 0.326], $p < .001$). There were also non-zero effects of attitudes ($H_8$; β = 0.182, 95% CI [0.076, 0.289], $p < .001$), subjective norms ($H_9$; β = 0.118, 95% CI [0.059, 0.178], $p < .001$), and perceived behavioral control ($H_{10}$; β = 0.324, 95% CI [0.228, 0.420], $p < .001$) on intentions. Moreover, there were non-zero effects of autonomous motivation in leisure time ($H_7$; β = 0.403, 95% CI [0.298, 0.507], $p < .001$) and trait self-control ($H_{12}$; β = 0.214, 95% CI [0.030, 0.399], $p = .012$) on intentions, but the effect of habit on intention ($H_{11}$) was no different from zero. We found non-zero effects of attitude ($H_{14}$; β = 0.093, 95% CI [0.034, 0.152], $p = .001$), trait self-control ($H_{17}$; β = 0.107, 95% CI [0.007, 0.207], $p = .018$), and past physical activity participation ($H_{22}$; β = 0.672, 95% CI [0.518, 0.827], $p < .001$) on leisure-time physical activity participation at follow-up, while effects of intention ($H_{13}$), perceived behavioral

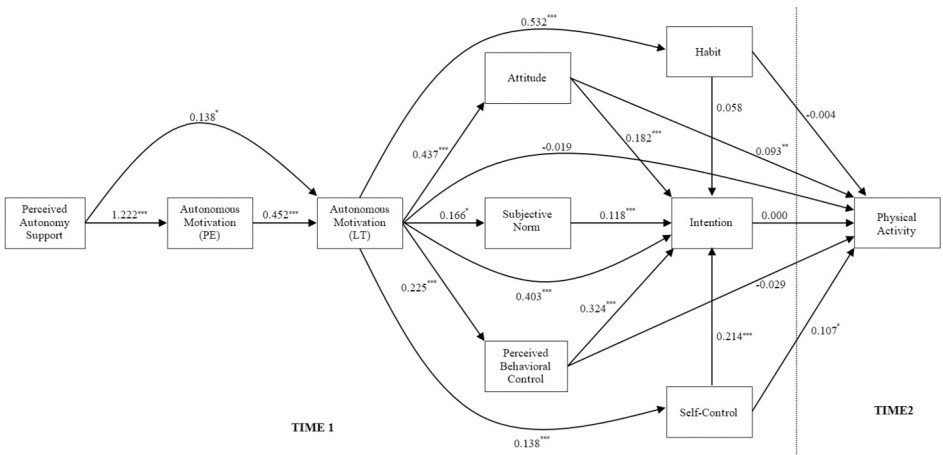

**Fig 2. Parameter estimates from the Bayesian path analysis of the extended trans-contextual model for leisure-time physical activity including informative priors.** PE = Physical education context; LT = Leisure time context. Model parameters omitted for clarity: past physical activity behavior→perceived autonomy support, β = 0.390, 95% CI [0.116, 0.664], *p* = .003; past physical activity behavior→autonomous motivation (PE), β = 0.368, 95% CI [-0.076, 0.799], *p* = .051; past physical activity behavior→autonomous motivation (LT), β = 1.382, 95% CI [1.109, 1.654], *p* < .001; past physical activity behavior→attitude, β = 0.194, 95% CI [-0.109, 0.495], *p* = .106; past physical activity behavior→subjective norms, β = 0.337, 95% CI [-0.193, 0.867], *p* = .103; past physical activity behavior→perceived behavioral control, β = 0.593, 95% CI [0.249, 0.938], *p* < .001; past physical activity behavior→intention, β = 0.293, 95% CI [0.014, 0.574], *p* = .019; past physical activity behavior→habit, β = 0.799, 95% CI [0.397, 1.206], *p* < .001; past physical activity behavior→self-control, β = 0.153, 95% CI [-0.041, 0.348], *p* = .061; past physical activity behavior→physical activity behavior, β = 0.672, 95% CI [0.518, 0.827], *p* = .001.

control ($H_{15}$), habit ($H_{16}$), and autonomous motivation in leisure time ($H_{18}$) were no different from zero. In addition, there were non-zero effects of autonomous motivation in leisure time on habit ($H_{19}$; β = 0.532, 95% CI [0.392, 0.671], *p* < .001) and trait self-control ($H_{20}$; β = 0.138, 95% CI [0.071, 0.207], *p* < .001). Finally, we found a non-zero effect of past behavior on habit ($H_{21}$; β = 0.799, 95% CI [0.397, 1.206], *p* < .001). All effects were small-to-medium in size.

Focusing on the indirect effects, we found non-zero indirect effects of perceived autonomy support on autonomous motivation in leisure time mediated by autonomous motivation in PE ($H_{23}$; β = 0.547, 95% CI [0.276, 0.777], *p* < .001). There were also non-zero effects of autonomous motivation in PE on intentions mediated by autonomous motivation in leisure time and attitude ($H_{24}$; β = 0.035, 95% CI [0.014, 0.061], *p* < .001), subjective norms ($H_{25}$; β = 0.008, 95% CI [0.001, 0.020], *p* = .017), and perceived behavioral control ($H_{26}$; β = 0.032, 95% CI [0.016, 0.054], *p* < .001). Similarly, there were also non-zero indirect effects of autonomous motivation in leisure time on intentions mediated by attitude ($H_{31}$; β = 0.079, 95% CI [0.032, 0.131], *p* < .001), subjective norms ($H_{32}$; β = 0.019, 95% CI [0.001, 0.044], *p* = .017), and perceived behavioral control ($H_{33}$; β = 0.072, 95% CI [0.037, 0.115], *p* < .001). There were also non-zero indirect effects of autonomous motivation in leisure-time on intention ($H_{35}$; β = 0.028, 95% CI [0.004, 0.064], *p* = .012) and physical activity participation ($H_{40}$; β = 0.014, 95% CI [0.001, 0.033], *p* = .018) mediated by trait self-control, but indirect effects mediated by habit were no different from zero ($H_{34}$, $H_{39}$). However, indirect effects of autonomous motivation in the PE ($H_{27}$–$H_{30}$) and leisure-time ($H_{36}$–$H_{38}$) contexts through the social cognition constructs on leisure-time physical activity participation were no different from zero, primarily because the intention-behavior relationship was also no different from zero. The indirect effect of past behavior on physical activity participation mediated by habit ($H_{41}$) was also no different from zero. Finally, effects of the intervention on leisure-time physical activity participation, and effects of age on model constructs, were no different from zero. However, we found non-zero

effects of gender on autonomous motivation in PE (β = -.401, 95% CI [-0.717, -0.099], *p* = .004), with girls experiencing higher levels than boys, autonomous motivation in leisure time (β = 0.256, 95% CI [0.055, 0.458], *p* = .007), and attitudes (β = 0.210, 95% CI [0.016, 0.402], *p* = .017).

## Discussion

The purpose of the current research was to examine the determinants of lower secondary school students' leisure-time physical activity participation using an extended version of the trans-contextual model [5]. Specifically, the model was augmented to include two constructs that reflect non-conscious processes as predictors of leisure-time physical activity participation: self-reported habit [21] and trait self-control [37]. In addition, attitude was also set as a direct predictor of leisure-time physical activity participation, representing a further non-conscious process [44, 45]. Hypothesized relations among the extended trans-contextual model constructs were tested using a two-wave prospective survey design in a convenience sample of lower secondary school students. Data were analyzed using two Bayesian path analytic models: one specifying non-informative priors and one in which informed priors for key relations in the model derived from previous research were specified. Results indicated adequate fit of both models with the data. Perceived autonomy support predicted autonomous motivation in PE and leisure-time contexts, autonomous motivation in PE predicted autonomous motivation in a leisure-time context, and autonomous motivation in a leisure-time context predicted social cognition constructs (attitudes, perceived behavioral control) and intentions toward leisure-time physical activity participation. There were also indirect effects of perceived autonomy support on autonomous motivation in leisure time mediated by autonomous motivation in PE, and of autonomous motivation in PE and leisure time on intentions through the social cognition constructs. In contrast, the hypothesized indirect effects of autonomous motivation in both contexts on leisure-time physical activity participation were not supported, primarily due to effects of intention and perceived behavioral control on behavior that were no different from zero. However, attitudes and trait self-control predicted both intentions and behavior. Furthermore, there were indirect effects of autonomous motivation in leisure-time on intentions and physical activity participation mediated by self-control, but not habit. The Bayesian analytic approach demonstrated that the model was tenable with the model incorporating informative prior knowledge demonstrating better fit with the data and more precision for some of the parameter estimates.

Overall, current results supported hypotheses relating to the first two premises of the trans-contextual model, that is, the premises specifying effects of perceived autonomy support on autonomous motivation in PE, and the trans-contextual effects of autonomous motivation across PE and leisure time context [5, 18]. It also provided support for the effects of autonomous motivation in leisure time on intentions to participate in leisure-time physical activity mediated by the attitude, subjective norm, and perceived behavioral control constructs from the theory of planned behavior. However, there was scant evidence for the third premise, due to an intention-physical activity participation relationship that was no different from zero. These findings suggest that, in the current sample, the trans-contextual model is effective in identifying motivational and social cognition determinants of secondary school students' intentions to participate in leisure-time physical activity, and the processes involved, but not their actual participation. We propose four possible interpretations of the current findings. First, results may raise questions on the effectiveness of the trans-contextual model in identifying the determinants of leisure-time physical activity participation. There have been occasions where studies on the motivational and social cognition constructs in multi-theory, integrated

models have failed to yield non-zero effects for the primary predicted determinants of behavior [17, 20, 66]. Nevertheless, such occasions are rare, and are contrary to the substantive body of meta-analytic evidence applying the trans-contextual model [18] and other integrated models that have supported effects more broadly and in multiple populations and contexts [67, 68]. Therefore, it may be premature to use the current data as a basis for rejecting the trans-contextual model.

A second interpretation may be that some of the hypothesized effects in the model were attenuated due to contextual factors that affected relations among constructs, particularly the intention-behavior relationship. It is important to note that the intention-behavior relationship is integral to the model as it is a key link in the 'motivational sequence' by which perceived autonomy support in PE and autonomous motivation in both contexts relates to leisure-time physical activity participation. An intention-behavior relationship that is no different from zero, therefore, suggests that the indirect effect of autonomy support and autonomous motivation in both contexts on behavior, a key premise of the model, is not supported. This should not, however, invalidate the model. Rather it may signpost potential contextual or environmental factors that lead to effects in the model are attenuated. For example, research has shown that extraneous constructs moderate the intention-behavior relationship [69].

One possibility is that the current research was conducted in the context of an intervention. However, correlations of the intervention with key model constructs, particularly intentions and follow-up physical activity participation were no different from zero. In fact, the only effects of the intervention on variables from the current study were on perceived autonomy support and attitudes at baseline, and these effects were opposite to the predicted direction and were taken prior to the intervention. Furthermore, we also controlled for intervention effects in the current model, so reported effects were independent of intervention effects. This leaves the possibility of other extraneous constructs attenuating the intention-physical activity participation relationship in the current study. It is possible, for example, that students' intentions were particularly unstable or inconsistent with their subsequent behavior, given research that has confirmed these intention properties moderate these relations [69]. However, this possibility remains speculative as we have no data on intention stability or consistency, nor do we have any contextual or demographic information that would explain such inconsistencies.

A third explanation may be that participation in leisure-time physical activity in the current sample of school students was largely determined by constructs that reflect individual-level non-conscious processes, that is, constructs that impact behavior directly independent of intentions. That the only determinants of leisure-time physical activity participation in the current study were past physical activity participation, attitude, and trait self-control is consistent with this interpretation. Focusing first on the direct effect of trait self-control on behavior, this construct is proposed to reflect non-conscious processes insofar as those endorsing it are purported to exhibit adaptive self-regulatory skills that assist in pursuing goal-directed behaviors and help resist temptations to engage in alternative behaviors that may derail pursuit of the behavior [39, 43]. On the surface, such an effect implies that individuals applying such skills must engage in active, effortful decision making to ensure focus on the target behavior and manage distractions, a conscious process. This may be the case for behaviors with which the individual has little experience. However, where the individual has substantive experience and has engaged in such active deliberation over the management of the behavior and application of their skills, they are likely to have well-learned behavioral scripts or schemas stored in memory to manage distractions and maintain behavioral engagement, obviating the need for such conscious deliberation. This is consistent with research suggesting that individuals with good trait self-control are highly effective in managing their environment so as not to be encumbered by distractions and to ensure that the cues to their desired behavior are

omnipresent [38]. While this mechanistic explanation is speculative, it may explain the direct effect of trait self-control on behavior in the current model and provides justification to explore the role of this constructs within the trans-contextual model.

A fourth and final interpretation is that social environmental factors may have contributed to the weak intention-behavior relationship. The high availability of inactive highly-appealing pastimes available to young people (e.g., computer games) and social norms within families and peer groups to engage in inactive pastimes may have contributed to failure of students to engage in physical activity even if they had autonomous motives and intentions to do so. This is consistent with the current data in which students' average intentions to engage in physical activity in their leisure time was above the scale mid-point ($M$ = 5.651, $SD$ = 1.282). The effects of peer norms are especially strong in this age group, so young people with intentions to be active may find that they are superseded by their need to conform. These premises are consistent with ecological models that stress environmental influences [e.g., 70], and research suggesting that such influences are important predictors of behavior beyond social cognition determinants [e.g., 71]. Analogously, if a child has low or no intention to participate in physical activity, they may still be compelled to spontaneously do so if their peer groups decides to have a 'kick about' with a football in their local park. The current study did not measure environmental influences, so such determinants cannot be empirically verified from the current data and should be considered speculative. Nevertheless, it points to the potential importance of incorporating constructs that reflect these environmental determinants within integrated models such as the trans-contextual model.

Turning to the direct effect of attitude on leisure-time physical activity participation, current findings are consistent with previous research that has found a direct effect of attitude components on behavior in multiple health contexts [44, 45]. Such effects might represent affective motives to engage in a behavior learned through positive or negative experiences that coincide with the behavior [72]. As a consequence, the anticipation of rewarding affective responses may be reasons why children and adolescents engage in physical activities outside of school without the need for reasoned decision making. Such an effect has not been identified in previous research adopting the trans-contextual model, but has been consistently identified in research applying the theory of planned behavior in health behavior contexts, including physical activity [45, 72].

With respect to the direct effect of past physical activity behavior, current findings corroborate previous research reporting effects of past behavior on subsequent behavior in social cognition theories [e.g., 14, 25, 47, 73]. This research demonstrates that past behavior accounts for substantive variance in behavior and often attenuates effects of other constructs. The inclusion of past behavior in social cognition models is important as it provides an indication of the sufficiency of the theory [9, 73]. The absence of effects of theory constructs other than past behavior provides an indication that the theory may be inadequate as a means to explain behavior beyond the stability of the behavior itself. Although in the case of the current research, the exclusion of past behavior did not restore effects of other constructs such as intention on behavior.

So, what might the large-sized effect of past behavior represent? Researchers have suggested that past behavior may model effects of unmeasured constructs in tests of these theories [9, 25]. Given social cognition theories incorporate constructs that reflect reasoned, deliberative processes, past behavior effects may model effects of constructs representing non-conscious processes such as habits and implicit beliefs. The substantive effect of past physical activity behavior on leisure-time physical activity participation in the current study suggests that lower secondary school students' physical activity in their leisure time may be a function of these kinds of constructs. Current findings suggest, however, that habit may not be among these

determinants, given that the independent effect of self-reported habit on leisure-time physical activity participation was no different from zero, and habit did not mediate effects of past behavior on physical activity participation. Although it must be stressed that the current measure of habit focused exclusively on automaticity, one aspect of habit, and may not have sufficiently captured all habitual influences [e.g., 74]. The current study did not include measures that capture other aspects of habit such as context stability and accessibility of relevant cues to the behavior [26]. In addition, we did not measure other constructs that may reflect these non-conscious processes, such as implicitly held beliefs developed through past experiences of the behavior covarying with evaluations [24]. Research has suggested that measures of implicit beliefs predict behavior, including physical activity participation, independent of intentions [20] and may also mediate effects of past behavior on subsequent behavior [28]. The effects of past behavior in the current study may, therefore, indicate that physical activity behavior in leisure time may be a function of unmeasured constructs reflecting implicit processes, but such an inference is speculative and requires empirical verification.

Finally, consistent with previous research on self-determination theory, we found an indirect effect of autonomous motivation in leisure-time on both intentions and behavior mediated by trait self-control. Research on self-determination theory suggests that autonomous motivation is associated with better self-regulatory capacity and resilience in the face of self-control resource depletion [40, 42]. These findings are an important augmentation of trans-contextual model as they provide an alternative process by which individuals enact leisure-time physical activity in the absence of the 'motivational sequence' outlined in the original model. This finding lends additional support to the 'energizing' effect of autonomous motivation–students in the current study were more likely to report greater self-control if they perceived their behavior to be autonomous. As self-control was measured as a trait and was not specific to physical activity in the current study, the self-control-autonomous motivation relationship in the current study may, in fact, reflect a general tendency for autonomously motivated individuals to report greater self-control. This needs to be corroborated at the trait level, such as examining relations between causality orientations from self-determination theory and trait self-control, and examine whether such individual differences are behaviorally relevant [75].

The current research also illustrates the value of adopting a Bayesian analytic approach to combine prior knowledge of the distributions of model effects with the observed distributions to produce precise estimates and variability among model constructs. This was demonstrated by the narrowing of the credibility intervals about some of the model parameters. Importantly, the data used for the informative priors was highly reliable given they were derived from meta-analyses of multiple studies with large samples sizes. It is, however, also important to note that although the informative priors for the trans-contextual model effects were a meta-analysis of studies on samples of school students with similar profile to the participants in the current study [18], priors for the effects of the additional variables, self-reported habit and trait self-control were derived from research from multiple populations and mostly adult samples [38, 46]. Therefore, the priors were not directly comparable to the current sample. Nevertheless, current findings may be of value as a source of informative priors for future applications of the extended trans-contextual model. Consistent with the Bayesian approach, the current study should form part of an ongoing iterative research process that yields increasingly precise estimates of effects in the model.

## Strengths, limitations and recommendations for future research

Strengths of the current study include (1) a focus on the determinants of lower secondary school students' leisure-time physical activity participation, a priority area of research; (2) the

application of an extended trans-contextual model, an integrative multi-theory approach that provided a priori hypotheses on the relations among the determinants and leisure-time physical activity participation; (3) adoption of a two-wave prospective design using validated measures of model determinants and behavior; and (4) application of Bayesian analytic procedures that enabled utilization of prior knowledge to arrive at precise estimates of model effects. However, it is also important to note limitations of the current research that may affect interpretation of the findings and the extent to which they can be generalized.

While we endeavored to incorporate additional constructs representing non-conscious determinants of leisure-time physical activity participation in the current study, our measures did not encompass a full range of candidate determinants. For example, the current study did not include measures of implicit cognition and motivation with respect to school students' leisure-time physical activity participation. Given that measures of constructs such as implicit beliefs and autonomous motivation have been shown to predict behavior directly independent of intentions in adult samples [20, 28, 76], future tests of the extended trans-contextual model should consider incorporating measures of these constructs as predictors of leisure-time physical activity participation. This is particularly important given the lack of effects of the intentional or motivational constructs on leisure-time physical activity participation in the current study, and inclusion of implicit beliefs may assist in providing an explanation of the effects of past behavior.

We also did not include the beliefs that underpin the attitude and subjective norm constructs [9]. Their effects on intentions and behavior are typically mediated by the direct attitude and subjective norm measures. Similarly, we did not include constructs related to socioecological environment that may determine behavior, and whose effects on behavior may be mediated by the social cognition constructs in the model [70]. There is precedence for the indirect effect of these beliefs and socio-ecological constructs in the model. Research has demonstrated that beliefs and socio-ecological factors relating to context and environment are related to the social cognition constructs that predict health behavior, and those constructs mediate the effects of the beliefs and socio-ecological factors on behavior [77, 78]. While the constructs in the current model are proposed to account for the effects of these variables, such influences need empirical verification and serve as an avenue for future research.

In addition, current data are correlational, which limits the extent to which we could infer causal relations among the extended trans-contextual model constructs. As with many model tests, including those of the trans-contextual model, causal effects are inferred from theory not the data [18]. Future research should consider the adoption of panel designs that permit modeling of temporal change and direction among trans-contextual model constructs over time through cross-lagged effects [12]. Such designs should also consider examining measuring model constructs over longer periods of time to test the capacity of the model to account for long-term change in its constructs and physical activity behavior, see Jacobs et al. [79] for an example. In addition, intervention and experimental designs that adopt appropriate behavior change techniques [80, 81] are needed to test the effect of manipulating the constructs found to have a direct effect on leisure-time physical activity participation [82]. For example, interventions targeting attitudes should seek to promote enjoyment and positive affect through positive experiences of physical activity, and interventions targeting self-discipline should seek to provide self-regulatory skills that promote better control over impulses to spend excessive time on leisure-time alternatives to physical activity (e.g., video games, watching television) and identify and manage barriers.

## Conclusion

The current research is the first to test an extended version of the trans-contextual model to identify determinants of leisure-time physical activity participation in lower secondary school

students. Results indicate that the traditional motivational and social cognition constructs are effective in predicting leisure-time physical activity intentions, but not actual behavior. However, we found direct effects of trait self-control and attitude on leisure-time physical activity participation, suggesting that students' physical activity participation was determined by constructs representing non-conscious processes. A further innovation of the current research is the application of a Bayesian analytic approach to update the effects and variability estimates of model parameters based on previous meta-analytic findings. Results raise questions over the effectiveness of the original trans-contextual model constructs in determining leisure-time physical activity participation, at least in the current societal context in which the physical environment may not support engagement in physical activity and offers various competing, non-active alternatives (e.g., video games). However, current findings highlight the potential of including additional constructs representing non-conscious processes. However, these data should not be considered unequivocal evidence to support rejection of the model as unmeasured moderator variables may have affected model effects. Further replication of the extended trans-contextual model predictions in larger samples is warranted.

## Supporting information

**S1 File.**
(DOCX)

## Acknowledgments

We thank Pauliina Hietanen, Noora Kilpeläinen, Sampsa Löppönen, Stina Seppänen, and Miika Tuominen for their assistance with data collection. We are grateful to Jari Villberg for his assistance with the data analysis. Jekaterina Schneider is now with the Centre for Appearance Research at University of the West of England, Bristol, UK.

## Author Contributions

**Conceptualization:** Juho Polet, Mary Hassandra, Taru Lintunen, Nelli Hankonen, Mirja Hirvensalo, Tuija H. Tammelin, Kyra Hamilton, Martin S. Hagger.

**Data curation:** Juho Polet, Jekaterina Schneider, Arto Laukkanen, Martin S. Hagger.

**Formal analysis:** Martin S. Hagger.

**Funding acquisition:** Mary Hassandra, Taru Lintunen, Nelli Hankonen, Mirja Hirvensalo, Tuija H. Tammelin, Martin S. Hagger.

**Investigation:** Juho Polet, Jekaterina Schneider, Mary Hassandra, Arto Laukkanen.

**Methodology:** Juho Polet, Jekaterina Schneider, Mary Hassandra, Taru Lintunen, Arto Laukkanen, Nelli Hankonen, Mirja Hirvensalo, Tuija H. Tammelin, Kyra Hamilton, Martin S. Hagger.

**Project administration:** Juho Polet, Jekaterina Schneider, Mary Hassandra, Taru Lintunen, Arto Laukkanen, Martin S. Hagger.

**Writing – original draft:** Juho Polet, Taru Lintunen, Martin S. Hagger.

**Writing – review & editing:** Juho Polet, Jekaterina Schneider, Mary Hassandra, Taru Lintunen, Arto Laukkanen, Nelli Hankonen, Mirja Hirvensalo, Tuija H. Tammelin, Kyra Hamilton, Martin S. Hagger.

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
