## [Decision Letter · Decision Letter 0]

19 Mar 2021

PONE-D-21-02994

Predictors of School Students’ Leisure-Time Physical Activity: An Extended Trans-Contextual Model Using Bayesian Path Analysis

PLOS ONE

Dear Dr. Hagger,

Thank you for submitting your manuscript to PLOS ONE. After careful consideration, we feel that it has merit but does not fully meet PLOS ONE’s publication criteria as it currently stands. Therefore, we invite you to submit a revised version of the manuscript that addresses the points raised during the review process.

Two expert reviewers with a track record of publications in the TCM and physical activity have reviewed your manuscript and provided detailed comments. I concur with their evaluation and have decided that minor revisions are needed before your manuscript is considered for publication. 

We look forward to receiving your revised manuscript.

Kind regards,

Lambros Lazuras

Academic Editor

PLOS ONE

Journal Requirements:

4. Thank you for submitting the above manuscript to PLOS ONE. During our internal evaluation of the manuscript, we found significant text overlap between your submission and the following previously published work, of which you are an author.

- https://www.sciencedirect.com/science/article/abs/pii/S1041608018301468?via%3Dihub ("Motivational predictors of students' participation in out-of-school learning activities and academic attainment in science: An application of the trans-contextual model using Bayesian path analysis" by Hagger and Hamilton, 2018)

Please revise the manuscript to rephrase the duplicated text, cite your sources, and provide details as to how the current manuscript advances on previous work. Please note that further consideration is dependent on the submission of a manuscript that addresses these concerns about the overlap in text with published work.

Reviewers' comments:

Reviewer's Responses to Questions

**Comments to the Author**

1. Is the manuscript technically sound, and do the data support the conclusions?

Reviewer #1: Yes

Reviewer #2: Yes

2. Has the statistical analysis been performed appropriately and rigorously? 

Reviewer #1: Yes

Reviewer #2: Yes

3. Have the authors made all data underlying the findings in their manuscript fully available?

Reviewer #1: Yes

Reviewer #2: Yes

4. Is the manuscript presented in an intelligible fashion and written in standard English?

Reviewer #1: Yes

Reviewer #2: Yes

5. Review Comments to the Author

Reviewer #1: General comments

This is an interesting manuscript testing an extended trans-contextual model to examine effects of motivational and social cognition constructs alongside the potential influence of implicit processes in influencing leisure time physical activity behavior among secondary school students. The study is based on sound theory with extended and clear theoretical explanations of the psychological mechanisms tested starting from the influence of PE teachers’ behavior on the psychological mechanisms operating to influence intentions and involvement in leisure time physical activity. Other positive aspects of the manuscript involve the prospective design used with behavior measured five weeks after assessment of the psychological constructs, the Bayesian method of path analyses conducted, and the testing of implicit processes added to the model. Overall, the study is well designed, clearly described in terms of the theoretical processes tested and the instruments used, statistical analysis is clearly reported, and the discussion covers in a satisfactory way study findings while the authors also state study limitations with sufficient analysis. Overall, I have been delighted reading the current manuscript and I believe it adds significant knowledge in the extant literature on motivational processes among secondary school students’ participation in leisure time physical activity.

Specific comments

Measures, Results. The authors have not reported confirmatory factor analysis results for the measurement tools used to measures the variables of the theory of planned behavior, habit, and trait self-control or other evidence supportive of the validity of scores derived from the instruments used in the present study. Therefore, it would strengthen the manuscript if the above information was presently added.

Reviewer #2: The present study investigated an extension of the trans-contextual

model (TCM) of motivation incorporating constructs related to implicit

process. It is a very interesting paper, technically well-written

providing sufficient contribution to the understanding of the effect

of school physical education on out-of-school physical activity. Below

there are some minor comments for authors to further increase the

clarity of their argumentation.

In the proposed model, the implicit processes constructs are not

somehow related to the TCM. In my understanding the core of the TCM is

that influences through a sequence of effects out-of-school physical

activity. In previous extensions of the model, the added variables

were clearly affected by autonomy supportive climate in physical

education and assisted in explaining more variance or introduced a new

pathway through which physical education climate influences

out-of-school physical activity. In the present study, this is not

apparent. The extensions of the model seem to work outside the core

hypotheses of the model. In fact, it seems that the added variables

address the intention-behavior gap, rather than contribute to the

motivational sequence proposed by the model. In this sense, it is not

clear to me how the proposed extension of the model helps us

understand the effect of physical education motivational climate on

out-of-school physical activity, which is the core aim of TCM (at

least as it is applied in the physical education-leisure time

association). In this sense, I would expect to see paths linking the

school environment (i.e., climate, motivational regulations) to

constructs such as trait self-control; there is evidence in the

literature linking autonomous motivation to self-regulation and

metacognition for instance. In this line, I would expect to see a path

from leisure-time autonomous motivation on habits; i.e., autonomously

motivated students would endorse more often habitual physical activity

or hold relevant beliefs.

Personally, I don’t see the lack of effect of intention on behavior as

a limitation of the TCM. The core and unique in the literature

premises of the model (i.e., trans-contextual effects of motivation,

effect of motivation on proximal to behavior social cognitive

variables) have been supported in the present study. The

intention-behavior gap, in my mind, is a limitation of the TPB and

had has been extensively studied in the literature. In this sense, I

don’t think that the authors should discard the value of TCM, as it

seems to be done in the discussion section.

In this line, I think that the second and third explanations offered

focus on the intention-behavior gap, rather than the utility of the

TCM. If the focus of the paper is on better understanding the

intention-behavior gap, then what is the value of testing the whole TCM?

Also, I am not sure I agree with the arguments presented in p. 26

about the spontaneous participation of children in physical activity.

And the reason for this is that if children hold positive views about

physical activity they will be actively involve in systematic physical

activity. If not, then spontaneous participation may be for external

to physical activity reasons, i.e., pressure by peers etc. In this

case the value of participation is questionable. I am not aware of any

evidence suggesting that such spontaneous participation in physical

activity will eventually result in more systematic participation in

the future. In this sense, the whole concept of implicit processes

should be conceived with caution about its contribution in

understanding systematic physical activity.

I found really interesting the discussion on the association of past

behavior and habits. Drawing back to my previous comment, I would

expect a path linking these two variables. This association may

provide, in my mind, a perfect bridge between explicit and implicit

processes associated with physical activity.

Overall, I think that a stronger association is needed of the added

variables with the core model.

6. PLOS authors have the option to publish the peer review history of their article (what does this mean?). If published, this will include your full peer review and any attached files.

Reviewer #1: No

Reviewer #2: No

---

## [Author Response · Author response to Decision Letter 0]

21 May 2021

RESPONSES TO REVIEWER 1’S COMMENTS

REVIEWER’S COMMENT 1: General comments

This is an interesting manuscript testing an extended trans-contextual model to examine effects of motivational and social cognition constructs alongside the potential influence of implicit processes in influencing leisure time physical activity behavior among secondary school students. The study is based on sound theory with extended and clear theoretical explanations of the psychological mechanisms tested starting from the influence of PE teachers’ behavior on the psychological mechanisms operating to influence intentions and involvement in leisure time physical activity. Other positive aspects of the manuscript involve the prospective design used with behavior measured five weeks after assessment of the psychological constructs, the Bayesian method of path analyses conducted, and the testing of implicit processes added to the model. Overall, the study is well designed, clearly described in terms of the theoretical processes tested and the instruments used, statistical analysis is clearly reported, and the discussion covers in a satisfactory way study findings while the authors also state study limitations with sufficient analysis. Overall, I have been delighted reading the current manuscript and I believe it adds significant knowledge in the extant literature on motivational processes among secondary school students’ participation in leisure time physical activity.

AUTHORS’ RESPONSE: We thank the Reviewer for the positive feedback on our manuscript and for highlighting its strengths and contribution. We have responded to their comments below, and have used the tracked changes function to mark the changes we have made to the revised manuscript.

REVIEWER’S COMMENT 2: Specific comments

Measures, Results. The authors have not reported confirmatory factor analysis results for the measurement tools used to measures the variables of the theory of planned behavior, habit, and trait self-control or other evidence supportive of the validity of scores derived from the instruments used in the present study. Therefore, it would strengthen the manuscript if the above information was presently added.

AUTHORS’ RESPONSE: The Reviewer raises an important point regarding the importance of confirmatory evidence for validity of the measures used to tap the model constructs. We did not conduct a full structural equation model in the current analysis because the complexity of the model relative to the sample size would have presented problems with convergence and stability of the factors, so such a model was contraindicated in this context. We did not conduct a confirmatory factor analysis for the same reasons. However, all measures were adapted directly from previous analyses using the trans-contextual model and the measures of habit and self-control have also been validated in numerous previous analyses. These analyses lend support for the construct validity of these measures through confirmatory factor analyses (e.g., factor loadings, average variance extracted, composite reliability). In addition, we conducted pilot analyses to test the validity of the measures used in the target population. These analyses used a single-indicator latent variable approach using omega reliability estimates to control for measurement error (note: simulation research using full structural equation models and single-indicator models has revealed little difference in the parameter estimates in models in either approach). These data provided support for the use of these measures in this population, particularly the predictive validity of the measures in the context of the trans-contextual model. We have cited these sources of validity in the revised manuscript (please see pages 13, lines 275-283 of the revised manuscript).

RESPONSES TO REVIEWER 2’S COMMENTS

REVIEWER’S COMMENT 1: The present study investigated an extension of the trans-contextual model (TCM) of motivation incorporating constructs related to implicit process. It is a very interesting paper, technically well-written providing sufficient contribution to the understanding of the effect of school physical education on out-of-school physical activity. Below there are some minor comments for authors to further increase the clarity of their argumentation.

AUTHORS’ RESPONSE: We are grateful to the Reviewer for providing positive feedback and constructive comments on our manuscript. We have responded to each comment in the point-by-point list below, and have used the track changes function to identify the changes we have made as a result of the revision process.

REVIEWER’S COMMENT 2: In the proposed model, the implicit processes constructs are not somehow related to the TCM. In my understanding the core of the TCM is that influences through a sequence of effects out-of-school physical activity. In previous extensions of the model, the added variables were clearly affected by autonomy supportive climate in physical education and assisted in explaining more variance or introduced a new pathway through which physical education climate influences out-of-school physical activity. In the present study, this is not apparent. The extensions of the model seem to work outside the core hypotheses of the model. In fact, it seems that the added variables address the intention-behavior gap, rather than contribute to the motivational sequence proposed by the model. In this sense, it is not clear to me how the proposed extension of the model helps us understand the effect of physical education motivational climate on out-of school physical activity, which is the core aim of TCM (at least as it is applied in the physical education-leisure time association). In this sense, I would expect to see paths linking the school environment (i.e., climate, motivational regulations) to constructs such as trait self-control; there is evidence in the literature linking autonomous motivation to self-regulation and metacognition for instance. In this line, I would expect to see a path from leisure-time autonomous motivation on habits; i.e., autonomously motivated students would endorse more often habitual physical activity or hold relevant beliefs.

AUTHORS’ RESPONSE: We thank the Reviewer for identifying this important issue. We agree that the original proposition of the TCM was to examine the processes by which autonomy support in PE relates to autonomous motivation toward physical activity in leisure time and intention toward, and actual participation in, leisure time physical activity. It does so by integrating constructs and hypotheses from self-determination theory (SDT) and the theory of planned behavior (TPB). The intention-behavior relationship, and the limitations thereof, are, as a consequence, integral to these processes. Knowledge of factors that may facilitate or inhibit the enactment of intentions, is, therefore, also integral to the TCM, so any constructs that may affect this relationship is also important to the model. In addition, the ultimate goal of the TCM is to explain variance in students’ leisure-time physical activity. As a consequence, augmenting the model to include constructs that represent other, parallel processes that may explain variance in physical activity in this population make important additions to the model and assist in its explanatory value. This is consistent with the suggestions made by Ajzen (1991) who indicated that the TPB is open to the inclusion of other constructs provided they make an independent contribution to the explanation of behavior. Other theorists, particularly those who have concerned themselves with the integration of theories (e.g., Hagger & Hamilton, 2020; Montaño & Kasprzyk, 2015), have made similar suggestions and highlighted that theories should be regarded as ‘living things’ and ‘works in progress’ that are not axiomatic or immutable, but open to modification and expansion provided those additions have conceptual value and, as Ajzen proposed, stand up to empirical rigor in facilitating explanation of behavior. The modified version of the TCM satisfies these criteria. The added constructs (habit and trait self-control) are consistent theoretically in that they outline alternative processes by which students may enact leisure-time physical activity, independent of the process outlined in the original model. These implicit processes are those that have been outlined in other social cognition models that have the TPB at their roots and previous integrated models (e.g., Hagger et al., 2017; Hamilton et al., 2020; van Bree et al., 2015). On the issue of relations between the motivational constructs from SDT and the added habit and self-control constructs in the extended model, we originally included correlations between these constructs as free parameters in the model, although we did not specify hypothesized relations among them. However, we agree with the Reviewer that these represent interesting and important additions. So we have included hypotheses for direct effects of autonomous motivation in leisure time on both habit (H19) and self-control (H20), and included them in the model. We have also specified, consistent with the Reviewer’s comments, potential indirect effects for autonomous motivation on intentions and behavior through these variables (see hypotheses H34, H35 and H39, H40 in the revised manuscript). These have been included in Tables 1 and 2, and specified in Figures 1 and 2. We have also included a summary of the conceptual basis for these effects in the Introduction section (please see page 7, lines 134-148 and page 8, lines 163-176) and the hypothesis section at the end of the Introduction (please see page 10, lines 210-219), reported the Results for these effects (please see page 24, lines 438-441), and explained their relevance to the TCM in the Discussion section (please see pages 31-32, lines 609-624).

REVIEWER’S COMMENT 3: Personally, I don’t see the lack of effect of intention on behavior as a limitation of the TCM. The core and unique in the literature premises of the model (i.e., trans contextual effects of motivation, effect of motivation on proximal to behavior social cognitive variables) have been supported in the present study. The intention-behavior gap, in my mind, is a limitation of the TPB and had has been extensively studied in the literature. In this sense, I don’t think that the authors should discard the value of TCM, as it seems to be done in the discussion section.

AUTHORS’ RESPONSE: As we stated in our response to the previous comment, the intention-behavior ‘gap’ is a key prediction of the TPB. It is also integral to the ‘motivational sequence’ and third premise of the TCM (Hagger & Chatzisarantis, 2016). The intention-behavior relationship should not, therefore, be disregarded in the context of the TCM. As the Reviewer notes in the previous comment, the TCM is concerned, through the proposed ‘motivational sequence’ with the processes that relate to intentions toward, and participation in, physical activity in leisure time. The intention-behavior relation is integral to this sequence, and so its rejection therefore would lead to a rejection of the motivational sequence, at least as a means to explain the motivational processes that lead to behavior. Consideration of non-deliberative predictors of physical activity behavior within the TCM is, therefore, important because it identifies potential determinants of physical activity participation in situations where the traditional TCM components may ‘fail’ to predict behavior. This would mean that the TCM is better equipped to explain instances (e.g., populations, contexts) in which intentions do not predict behavior and the motivational sequence does not account for variance in the key dependent variable of the model. We have, however, been careful not to dismiss the TCM outright, but rather highlight that for certain populations and in certain contexts, the lack of association between intention and behavior may mean that alternative processes that act in parallel to the motivational sequence may be accounting for behavior (please see page 26-27, lines 492-500 of the revised manuscript).

REVIEWER’S COMMENT 4: In this line, I think that the second and third explanations offered focus on the intention-behavior gap, rather than the utility of the TCM. If the focus of the paper is on better understanding the intention-behavior gap, then what is the value of testing the whole TCM?

AUTHORS’ RESPONSE: As we outlined in our responses to the previous comments, the intention-behavior gap is integral to the motivational sequence of the TCM, and therefore any research on the TCM should be concerned with the intention-behavior relation because, if the association is small or no different from zero, as is the case in the current research, it means that the motivational sequence is not effective as a description of the processes that lead to leisure-time physical activity participation. Therefore, means to explain why this is the case in the context of the TCM adds value to the model when its traditional components do not provide an adequate account for variance in physical activity. Please see our changes to the manuscript, noted in our response to the comment above, with regard to this issue. We also provide additional discussion of the importance of the additions in the Discussion section on page 27, lines 503-510 of the revised manuscript.

REVIEWER’S COMMENT 5: Also, I am not sure I agree with the arguments presented in p. 26 about the spontaneous participation of children in physical activity. And the reason for this is that if children hold positive views about physical activity they will be actively involve in systematic physical activity. If not, then spontaneous participation may be for external to physical activity reasons, i.e., pressure by peers etc. In this case the value of participation is questionable. I am not aware of any evidence suggesting that such spontaneous participation in physical activity will eventually result in more systematic participation in the future. In this sense, the whole concept of implicit processes should be conceived with caution about its contribution in understanding systematic physical activity.

AUTHORS’ RESPONSE: We agree, and we have removed the reference to spontaneous participation of children in physical activity and reformulated the manuscript excerpt as follows: “Such effects might represent affective motives to engage in a behavior learned through positive or negative experiences that coincide with the behavior (Conroy & Berry, 2017). As a consequence, the anticipation of rewarding affective responses may be reasons why children and adolescents engage in physical activities outside of school without the need for reasoned decision making”. Please see our addition to the revised manuscript on pages 29-30, lines 568-575.

REVIEWER’S COMMENT 6: I found really interesting the discussion on the association of past behavior and habits. Drawing back to my previous comment, I would expect a path linking these two variables. This association may provide, in my mind, a perfect bridge between explicit and implicit processes associated with physical activity.

AUTHORS’ RESPONSE: Thanks for this comment. We appreciate the need to provide links between previous experience and habit, to demonstrate that performing the behavior frequently in the past informs whether participants reflect that their behavior experienced as ‘automatic’ and determined without extensive deliberation. In fact, we included past behavior as a predictor of habit in the model. For clarity, we did not illustrate this in Figure 2, but mention it in the figure caption. However, we agree that it may be important to specify this indirect effect as it sheds light on a key process in the model, that the link between past and future behavior is mediated by habit. So we have included this as an additional hypothesis in the model in the Introduction (please see page 10, lines 210-219) and also in Tables 1 and 2 (H41), the Results (please see page 25, lines 445-446, and Table 2), and Discussion (please see page 26, lines 475-477 and page 30, lines 593-596) sections of the revised manuscript.

REVIEWER’S COMMENT 7: Overall, I think that a stronger association is needed of the added variables with the core model.

AUTHORS’ RESPONSE: We thank the Reviewer for the previous comments. These justifications have now been provided, and we have noted the location of the additions in our responses to the above comments.

REFERENCES

Ajzen, I. (1991). The theory of planned behavior. Organizational Behavior and Human Decision Processes, 50(2), 179-211. https://doi.org/10.1016/0749-5978(91)90020-T

Hagger, M. S., & Chatzisarantis, N. L. D. (2016). The trans-contextual model of autonomous motivation in education: Conceptual and empirical issues and meta-analysis. Review of Educational Research, 86(2), 360-407. https://doi.org/10.3102/0034654315585005

Hagger, M. S., & Hamilton, K. (2020). Changing behavior using integrated theories. In M. S. Hagger, L. D. Cameron, K. Hamilton, N. Hankonen & T. Lintunen (Eds.), The Handbook of Behavior Change (pp. 208-224). Cambridge University Press. https://doi.org/10.1017/97811086773180.015

Hagger, M. S., Trost, N., Keech, J., Chan, D. K. C., & Hamilton, K. (2017). Predicting sugar consumption: Application of an integrated dual-process, dual-phase model. Appetite, 116, 147-156. https://doi.org/10.1016/j.appet.2017.04.032

Hamilton, K., Gibbs, I., Keech, J. J., & Hagger, M. S. (2020). Reasoned and implicit processes in heavy episodic drinking: An integrated dual process model. British Journal of Health Psychology, 25(1), 189-209. https://doi.org/10.1111/BJHP.12401

Montaño, D. E., & Kasprzyk, D. (2015). Theory of reasoned action, theory of planned behavior, and the integrated behavioral model. In K. Glanz, B. K. Rimer & K. Viswanath (Eds.), Health behavior and health education: Theory, research, and practice (5th ed., pp. 95-124). Jossey-Bass. 

van Bree, R. J. H., van Stralen, M. M., Mudde, A. N., Bolman, C., de Vries, H., & Lechner, L. (2015). Habit as mediator of the relationship between prior and later physical activity: A longitudinal study in older adults. Psychology of Sport and Exercise, 19(1), 95-102. https://doi.org/10.1016/j.psychsport.2015.03.006

---

## [Decision Letter · Decision Letter 1]

7 Oct 2021

Predictors of School Students’ Leisure-Time Physical Activity: An Extended Trans-Contextual Model Using Bayesian Path Analysis

PONE-D-21-02994R1

Dear Dr. Hagger,

We’re pleased to inform you that your manuscript has been judged scientifically suitable for publication and will be formally accepted for publication once it meets all outstanding technical requirements.

Kind regards,

Lambros Lazuras

Academic Editor

PLOS ONE

Additional Editor Comments (optional):

Reviewers' comments:

Reviewer's Responses to Questions

**Comments to the Author**

1. If the authors have adequately addressed your comments raised in a previous round of review and you feel that this manuscript is now acceptable for publication, you may indicate that here to bypass the “Comments to the Author” section, enter your conflict of interest statement in the “Confidential to Editor” section, and submit your "Accept" recommendation.

Reviewer #1: All comments have been addressed

Reviewer #2: All comments have been addressed

2. Is the manuscript technically sound, and do the data support the conclusions?

Reviewer #1: Yes

Reviewer #2: Yes

3. Has the statistical analysis been performed appropriately and rigorously? 

Reviewer #1: Yes

Reviewer #2: Yes

4. Have the authors made all data underlying the findings in their manuscript fully available?

Reviewer #1: Yes

Reviewer #2: Yes

5. Is the manuscript presented in an intelligible fashion and written in standard English?

Reviewer #1: Yes

Reviewer #2: Yes

6. Review Comments to the Author

Reviewer #1: (No Response)

Reviewer #2: (No Response)

7. PLOS authors have the option to publish the peer review history of their article (what does this mean?). If published, this will include your full peer review and any attached files.

Reviewer #1: No

Reviewer #2: No

---

## [Editor Report · Acceptance letter]

26 Oct 2021

PONE-D-21-02994R1 

Predictors of School Students’ Leisure-Time Physical Activity: An Extended Trans-Contextual Model Using Bayesian Path Analysis 

Dear Dr. Hagger:

I'm pleased to inform you that your manuscript has been deemed suitable for publication in PLOS ONE. Congratulations! Your manuscript is now with our production department. 

Kind regards, 

on behalf of

Dr. Lambros Lazuras 

Academic Editor

PLOS ONE